

# Characteristic Analysis of the Differences between TEC Values in GIM Grids

Qisheng Wang[1], Jiaru Zhu[1]

[1]College of Civil Engineering, Xiangtan University, Xiangtan 411105, China

*Correspondence to*: Qisheng Wang (qswang@xtu.edu.cn)

**Abstract.** Using total electron content (TEC) from global ionosphere map (GIM) for ionospheric delay correction is a common method of eliminating ionospheric errors in satellite navigation and positioning. On this basis, the TEC of puncture point can be obtained by GIM grid TEC interpolation. However, in terms of grid, only few studies have analyzed the TEC

value size characteristics of its four grid points, that is, the TEC difference characteristics among them. In view of this, by utilizing the GIM data from high solar activity years (2014) and low solar activity years (2021) provided by CODE, this paper proposes the grid TEC difference to analyze TEC variation characteristics within the grid, which is conducive to exploring and analyzing the variation characteristics of the ionosphere TEC in the single-station area. The results show that the TEC difference size within GIM grid is mainly related to the activity of ionosphere. The value is larger in high solar

activity years and generally small in low solar activity years, and the value of high latitude area is always smaller than that of low latitude area. Specifically, in high solar activity years, most of the GIM grid TEC internal differences are within 4TECu in high and mid latitude regions, while only 78.17% in low latitude regions; the grid TEC differences at 2-hour intervals are more scattered, and larger values occur in low latitude regions. In low solar activity years, the TEC difference values within GIM grid are mostly less than 2TECu, and most of them in the high and middle latitudes are within 1TECu. The GIM grid

TEC difference values within 1-hour intervals are mostly less than 4TECu, and most of them in the high and middle latitudes are within 2TECu. The main finding of this analysis is that the grid TEC differences are small for most GIM grids, especially in the mid-high latitudes of low solar years. This means that relevant extraction methods and processes can be simplified when TEC within these GIM grids is needed.

## 1 Introduction

Ionospheric delay is an important error source in navigation, positioning and timing of Global Navigation Satellite System (GNSS) (Hernández-Pajares et al., 2018;Hu et al., 2018;Jin et al., 2015), which affects the accuracy of GNSS on the one hand. And on the other hand, global all-weather observations of GNSS can be fully used to construct a global ionospheric model(Chen et al., 2020;Hernández-Pajares et al., 2009;Hernández-Pajares et al., 2011). Combined with total electron content (TEC) parameterized by ionospheric delay, the global ionosphere map (GIM) can be generated by TEC modeling





based on the globally distributed GNSS observations(Mannucci et al., 1998;Schaer, 1999;Hernández-Pajares et al., 2017;Zhang and Zhao, 2018). GIM can be mainly applied in the following fields: (1) The TEC provided by GIM for ionospheric delay correction is a common method to eliminate ionospheric errors in satellite navigation and positioning(Rovira-Garcia et al., 2019;Su et al., 2019); (2) GIM can be employed to eliminate TEC parameters in GNSS observation equations, thereby obtaining the code bias parameters of satellites and receivers(Montenbruck et al., 2014;Li et al., 2017); (3) GIM can be adopted to analyze and study the characteristics of global or regional ionospheric variations(Feng et al., 2022;Feng et al., 2023). It should be mentioned that the above applications need to focus on the grid TEC information. For example, when performing ionospheric delay correction, the TEC value of the puncture point needs to be obtained by interpolating the TEC of the gridded grid where the puncture point is located(Jin et al., 2012). Therefore, taking the GIM grid as an object, it is meaningful to analyze the variation of TEC difference within the grid, which further facilitates more in-depth understanding of the variation characteristics of the ionosphere in the single-station region.

Since the ionosphere is influenced by solar activity, its system state and variation are complicated, but it is generally believed that the active level of the ionosphere is related to solar activity. A number of studies worldwide have demonstrated that the ionosphere exhibits equatorial anomalies and latitudinal effects in space, and meanwhile periodic variations with the high and low solar activity in time(Tariq et al., 2020;Muafiry et al., 2022;Yu et al., 2014;Kalinin and Khotenko, 2012). In addition, GIM has also been utilized to conduct relevant research on the spatiotemporal variation characteristics of the regional ionospheric TEC(Guo et al., 2017). However, most of the studies on the ionospheric TEC variation characteristics focus on large scale. Considering that the ionospheric penetration point region formed by GNSS observation at a single station may contain several adjacent grids, the characteristics of the ionospheric TEC variation in such a single station area are rarely analyzed, especially in grid units. Moreover, using TEC from GIM for ionospheric delay correction is a common method of eliminating ionospheric errors in satellite navigation and positioning. With the aid of the method, the TEC of puncture point can be obtained by GIM grid TEC interpolation. However, in terms of grids, few studies have been performed to analyze the TEC difference characteristics of its four grid points. Hence, an accurate and comprehensive analysis of the variation of TEC difference in grid is of great importance, which is helpful to understand the variation characteristics of the ionosphere in the single-station area.

Given this, the grid TEC difference is proposed to analyze TEC variation characteristics within the grid. The GIM data of two years from high solar activity (2014) and low solar activity (2021) provided by CODE are selected to calculate the TEC difference for each grid point in this paper. Based on the calculation of the spatial and temporal variations of the difference values, both spatial and temporal characteristics of the TEC difference values of the four grid points within the grid are analyzed in detail.

This paper is organized as follows. In Section 2, related methods and data are introduced, especially the definition and calculation of grid TEC differences. In Section 3, the spatial and temporal characteristics of the TEC difference values of the four grid points within the grid are analyzed. In Section 4, the significance of the analysis in this paper is further illustrated through discussions. Section 5 presents conclusions of this paper.



## 2. Method and Data

### 2.1. Grid TEC

The GIM provided by CODE (Center for Orbit Determination in Europe) plays an important role in ionospheric research. By using globally distributed IGS tracking stations, GIM can be employed to generate a grid TEC model by 5° longitude and 2.5° latitude by spherical harmonic function modeling. Earlier CODE's GIM was a grid map at 2-hour intervals, with a day divided into 13 maps, while the interval of the current GIM is 1 hour, with 25 maps per day.

Figure 1 describes a schematic diagram of GIM grids with 5° interval in longitude direction and 2.5° interval in latitude direction, each of which has four grid points (as shown in Fig. a, b, c, d), indicating that there are $70 \times 72$ grids and $71 \times 73$ grid points TEC values in each TEC map. The grid TEC described in this paper refers to the TEC value of a grid, which includes the TEC value of the four grid points and the TEC value inside the grid. In practice, the grid TEC value is variable, but the GIM-provided grid TEC has only four grid point values. It should be noted that the analysis in this paper is based on GIM, and does not consider the problem of low TEC accuracy in some areas due to uneven or insufficient GNSS tracking stations.

For a certain grid, the TEC of a certain point inside it is calculated by the four-grid points TEC of the grid, which is also known as interpolation calculation. Figure 2 shows the distribution of puncture points in ABPO (GNSS tracking station), It can be seen that It can be seen that the puncture points are in a certain grid. A certain puncture point is in a certain grid, and its TEC value is obtained by interpolating the TEC of the four grid points when using GIM for ionospheric delay correction or TEC elimination. Therefore, understanding the variation of TEC values in these grids not only can provide a theoretical reference for obtaining the TEC values of the puncture point, but also acquire the information of variation characteristics of the ionosphere in the area of a single GNSS station. In other words, it is to analyze the spatial and temporal variation characteristics of the four-grid point TECs of each grid.

### 2.2. Grid TEC difference

In this paper, the grid TEC difference is proposed aims to analyze the TEC variation characteristics within the grid. The grid TEC difference includes the difference on the spatial scale and the difference on the temporal scale. The former is defined as the difference between the four grid points of a grid, and the latter is defined as the difference between four grid points in the grid of two adjacent GIMs. Specifically, on the premise of treating these grids as units, through calculating the grid TEC difference values of each grid, the variation of TEC difference values within these grids in space and time are counted, and both spatial and temporal variation characteristics of TEC difference values of four grid points within the grids are analyzed.

On the spatial scale, the grid difference values of each GIM are firstly calculated as shown in Eq. (1), and the maximum, average and minimum values of TEC difference values of each grid are counted. Afterwards, the spatial variation





characteristics of grid difference values in different periods are analyzed. Finally, the variation pattern of TEC difference values of grid in a day is obtained. The grid TEC difference on the spatial scale can be expressed as:

$$\Delta T_{jk} = \left| T_j - T_k \right| \quad j = \{a,a,a,b,b,c\} \quad k = \{b,c,d,c,d,d\} \tag{1}$$

where $T_j$ and $T_k$ is the TEC of grid point $j$ and $k$, respectively ; $\Delta T_{jk}$ is the grid TEC difference; there are six $\Delta T$ each grid.

On the temporal scale, the grid difference values between adjacent moments of each GIM are calculated as shown in Eq. (2), and the maximum, average and minimum values of TEC difference values of each grid are also counted. Then, the temporal variation characteristics of grid difference values in different periods are analyzed. Finally, the variation pattern of TEC difference values of grid in a day is achieved. The grid TEC difference on a time scale can be expressed as:

$$\Delta T_j^n = \left| T_j^{n+1} - T_j^n \right| \quad j = \{a,b,c,d\} \tag{2}$$

where $T_j^n$ is the TEC of grid point $j$ for nth GIM map, n=12 in 2014 and n=24 in 2021, $\Delta T_j^n$ is the TEC difference of grid point between adjacent maps ; there are four $\Delta T$ each grid.

Since the TEC of the puncture point is interpolated through the grid point, it is crucial to analyze the variation of TEC difference in grid, which contributes to gaining insight into the variation characteristics of the ionosphere in the single-station area. Moreover, understanding the characteristics of TEC difference in grid can provide a simplified idea for

obtaining TEC at puncture points.

**2.3. Data**

The GIM produced by CODE analysis center is used as the analysis data for this paper. Considering that ionospheric changes are influenced by solar activity, the F10.7 index is utilized to reflect the degree of solar activity. The monthly average F10.7 index changes from 2010 to 2021 are collected, as shown in Figure 3. From the figure, it can be seen that the

highest F10.7 index in 2014 represented a high solar activity year.

In order to distinguish the TEC changes in high and low solar activity years, the GIM data of 2014 (high solar activity year) and 2021 (low solar activity year) are selected for analysis in this paper. There are 25 maps for October 19, 2014 and the day after, with 365 days in the year, and all GIM graphs with 2-hour interval are selected for the unified analysis of 2014. However, the 300th day of 2021 data file is corrupted, suggesting there are 364 days of data available for the year.

**3. Results and analysis**

**3.1. Spatial Variation**

The data of 2014 and 2021 are used to make differences between the grid TECs of each GIM, with six differences for each grid. In order to analyze the variation of the grid TEC difference, the GIM of an arbitrary day (DOY112) is selected,





and the maximum, mean and minimum values of the absolute values of the grid TEC difference are counted. The results of
the variation with latitude at four moments of the day, 02, 08, 14 and 20 are summarized in Figure 3. The maximum, mean
and minimum values are indicated by three colors, and their mean values are also marked on the graph by three colors. The
first and second rows denote the results for 2014 and 2021, respectively, and the first to fourth columns represent the results
for the four moments, respectively.

As can be seen from Figure 4, the maximum, mean and minimum values of the absolute value of the grid TEC
difference are larger and show a bimodal variation in the low-latitude region, i.e., the highest value is shown near 30° north-
south latitude and decreases near the equator, with relatively small values appearing between them. This is because of the
sudden increase in the TEC value of the grid point at 30 degrees in the GIM, resulting in a large grid TEC difference near 30
degrees north and south latitude. Although the TEC values of grid points between 30 degrees north and south latitudes are
large, their differences are small, leading to the bimodal phenomenon in the figure. The reason is that due to active variation
of ionosphere at low latitudes, its TEC shows large values, and the grid TEC difference increases abruptly near 30° north and
south latitude. Comparing the results of the two years, the grid TEC difference is larger in 2014. In the meantime, it is
evident from their mean values that the ionospheric variability is more active in high solar activity years, with larger
differences of the grid TEC exhibited. Through observation, all figures present the gradual increase in the variation of the
grid TEC difference from high to low latitudes, indicating that the variation of the grid TEC difference is closely associated
with the latitude at which the grid TEC is located.

Like Figure 5, the variation of GIM grid TEC difference in longitude shown in Figure 5. It is obvious that the change of
grid TEC difference has no obvious characteristics in the direction of longitude, which is different from Figure 3. This
indicates that the change of grid TEC difference has a certain relationship with latitude. Therefore, subsequent analyses are
mainly in the latitudinal direction.

To further analyze the variation of grid TEC differences over the year, the GIM grids are counted separately by high,
mid, and low latitudes, with 22, 24, and 24 grids per map, respectively. The maximum, mean and minimum values of grid
TEC differences are averaged over 13 or 25 GIMs of a day in high, middle and low latitudes. It should be noted that the
difference values of the statistics here are considered as absolute values. The results of the statistics are tabulated in Fig. 6,
where the maximum, mean and minimum values of TEC differences of the grid in 2014 and 2021 are indicated by 6 colors,
and their mean values for one year are also represented on the graph by different colors. From the figure, the maximum,
mean and minimum values of grid TEC differences in 2014 show obvious fluctuations in all three regions. Especially the
maximum and mean values increase and decrease twice, which may be related to ionospheric activities. This trend is the
same as the trend of F10.7 in 2014 in Figure 1. Nevertheless, F10.7 in 2021 displays a slower trend of variation. In the low
solar activity year, the maximum, mean and minimum values of the grid TEC difference in Fig. 4 also show a slower annual
variation trend. Among the three latitudes, the low latitudes are more active while both high and mid latitudes are relatively
flat. The daily average value of the maximum grid TEC difference is close to 8 TECu in 2014 and around 4 TECu in 2021,
while that value is within 4 TECu and 3 TECu in mid and high latitudes, respectively. For the minimum value of the gridded



TEC difference, the mean values are within 2TECu for both years. This indicates that the factors affecting the magnitude of the GIM grid TEC difference mainly include solar activity and the latitude at which the grid is located.

For further analysis, the TEC difference of all grids in a year is counted, and there are 6 differences for each grid. In 2014, all the grids are counted at 2-hour intervals of 13 maps a day, and there are 6×70×72×13×365=143488800 differences; in 2021, the grids are counted at 1-hour intervals of 25 maps a day, and there are 6×70×72×25×364= 275184000 differences. As in the previous section, the frequency of grid TEC differences between -8 and 8TECu at 2TECu intervals is still counted separately by high and low latitudes. The statistical results for 2014 and 2021 are organized in Tables 1 and 2. The

histograms of TEC grid differences by each of the three latitudes are depicted in Figure 5.

As can be seen from the results in Table 1, 72.11% of the 2014 GIM grid TEC differences are in the range of -2 to 2TECu, 87.75% of the grid TEC differences are in the range of -2 to 2TECu in the high latitude region, while the values of grid TEC differences account for 76.71% and 53.19% in mid-latitude and low-latitude regions, respectively. Moreover, 90.20% of the grid TEC difference are in the range of -4 to 4TECu for 2014 GIM, while the values of grid TEC differences

account for 98.38%, 94.73% and 78.17% in high, mid and low latitude regions, respectively. Obviously, the TEC difference values present a relatively larger variation trend in the low latitude region in 2014. This is attributed to more active ionospheric variability in a high solar activity year of 2014 in the low latitude region, which is also consistent with the results of the previous analysis.

From the results in Table 2, 93.69% of the GIM grid TEC difference values in 2021 are in the range of -2~2TECu.

99.61% of the grid TEC difference values in the range of -2~2TECu are in the high latitude region, while 97.22% and 84.72% of TEC difference values in the range of -2~2TECu are in mid-latitude and low latitude regions, respectively. Moreover, 98.99% of the GIM in 2021 for grid TEC differences are in the range of -4 to 4TECu, and 99.99%, 99.78%, and 97.26% for high, mid, and low latitude regions, respectively. Clearly, the range of GIM grid TEC difference is larger in the range of -2 to 2TECu in 2021 compared to 2014. Most of the grid TEC differences are less than 2TECu in low solar activity years like

2021, and almost all grid TEC differences are within 4TECu, which is related to the flattening of the ionospheric activity due to lower solar activity.

The histogram of grid TEC differences in Figure 7 also reveals that a larger proportion of high and mid-latitude regions have a smaller range of TEC differences than low-latitude regions, and a larger proportion of 2021 has a smaller range of TEC differences than 2014. In addition, the GIM grid TEC differences all obey normal distribution, and most of the grid

TEC differences are within a certain range, especially for the high latitude region in 2021. It can be found that most of its grid TEC differences are within 1TECu, and most of its mid-latitude region is also within 2TECu. In summary, the TEC differences of the GIM grid are smaller in the high and mid-latitude regions where the ionosphere changes slowly in low solar activity years.



### 3.2. Temporal variation

On the time scale, the adjacent moments of each GIM map are differenced, and there are 4 differences for each grid. By taking the difference of each grid as a unit, the maximum, mean and minimum values of the absolute values of these differences are counted to analyze the change of TEC of the grid in time, and then the change of TEC of the GIM grid in time for the whole year is counted. It should be noted that, for the sake of data uniformity, all GIMs are counted at 2-hour intervals, i.e., 13 frames per day, in 2014, and 25 frames per day at 1-hour intervals in 2021. In order to take into account the

effect of the latitude of the grid, these results still need to be counted separately for high, medium and low latitudes.

The daily average results of the maximum, average and minimum values of the grid TEC difference between the two GIMs at adjacent moments of each day in 2014 and 2021 are enumerated in Figure 8. It is obvious from the figure that the variation of the grid TEC difference in 2014 fluctuates greatly, and the trend of the fluctuation is consistent in the three latitudinal regions and the same as the variation of the F10.7 index. It is due to the high solar activity year and active

ionospheric variation in 2014, which conforms to the previous results. Despite the large variation of the grid TEC difference in 2014, it is still evident that the high and middle latitudes are smaller than the low latitudes specifically in terms of values. In contrast, the variation trend of the grid TEC difference in 2021 is relatively gentle and most of the variation values in high and low latitudes are within 2TECu. On the one hand, this is because 2021 is a low solar activity year with a flat ionospheric activity. And on the other hand, it is attributed to the GIM time interval of 1 hour in 2021, while the interval in 2014 is 2

hours.

For further analysis, the TEC differences of all adjacent time grids in a year are counted, and there are 4 differences for each grid. In 2014, all the grids are counted at 2-hour intervals of 13 maps a day, and there are 4×70×72×12×365=88300800 differences; in 2021, the grids are counted at 1-hour intervals of 25 maps a day, and there are 4×70×72×24×364= 176117760 differences. As in the previous section, the frequency of grid TEC differences between -8 and 8TECu at 2TECu intervals is

still counted separately by high and low latitudes. The statistical results for 2014 and 2021 are introduced in Table 3 and Table 4. The histograms of TEC grid differences by each of the three latitudes are drawn in Figure 9.

As can be seen in Table 3, the results of TEC difference values of the GIM grid at adjacent moments in 2014 are relatively scattered, with only 53.55% within 4TECu globally, only 74.66% of the values in the high latitude region, with a minimum of only 30.11% in the low latitude region, and with nearly half of the TEC difference values exceeding 8TECu in

the low latitude region, which is related to the active ionospheric changes during the high solar activity year of 2014. Table 4 provides the statistical results of TEC difference of GIM grid in adjacent time periods in 2021. Obviously, when the difference range is within 4TECu, it accounts for 93.63% globally, while the percentage of high and low latitude regions are 99.92%, 98.96% and 82.53%, respectively. In particular, 97.22% of the TEC difference values in high latitude regions account for less than 2TECu.

Figure 9 also gives the histograms of the TEC differences of the adjacent moment grids for the high and low latitude regions in 2014 and 2021, respectively. It can be clearly found that they abide by normal distribution, but there are some



differences in the ranges of their respective distributions. Furthermore, the distribution of the difference in 2014 has a larger range, especially in the low-latitude region with more than 20 TECu; while for 2021, most of its differences in the high-latitude region are within 2 TECu.

## 4. Conclusions

By utilizing the GIM data from high solar activity years (2014) and low solar activity years (2021) provided by CODE, this paper proposes the grid TEC difference to analyze TEC variation characteristics within the grid, which is conducive to exploring and analyzing the variation characteristics of the ionosphere TEC in the single-station area. The results show that the TEC difference size within GIM grid is mainly related to the activity of ionosphere. The value is larger in high solar activity years and generally small in low solar activity years, and the value of high latitude area is always smaller than that of low latitude area. Specifically, in high solar activity years, most of the GIM grid TEC internal differences are within 4TECu in high and mid latitude regions, while only 78.17% in low latitude regions; the grid TEC differences at 2-hour intervals are more scattered, and larger differences occur in low latitude regions. In low solar activity years, the TEC difference values within GIM grid are mostly less than 2TECu, and most of them in the high and middle latitudes are within 1TECu. The GIM grid TEC difference values within 1-hour intervals are mostly less than 4TECu, and most of them in the high and middle latitudes are within 2TECu. The main finding of this analysis is that the grid TEC differences are small for most GIM grids, especially in the mid-high latitudes of low solar years. This means that relevant extraction methods and processes can be simplified when TEC within these GIM grids is needed.

The results of the above analysis can help to understand the ionospheric TEC variation characteristics in the GNSS single-station region (which may be the range of several adjacent grids), and provide corresponding reference for regional ionospheric modeling. Especially for the high and mid-latitude regions with low solar activity years. Since the TEC difference within the grid varies less, and the TEC processes can be simplified accordingly in terms of GNSS single-frequency ionospheric delay correction, single-station regional ionospheric modeling and code bias estimation, etc. The related validation and analysis need to be further studied.

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

**Table 1 Statistics of GIM grid TEC difference in 2014**

| Latitude range | Global | | High-latitude | | Mid-latitude | | Low-latitude | |
|---|---|---|---|---|---|---|---|---|
| TECu | Number | Percentage | Number | Percentage | Number | Percentage | Number | Percentage |
| <-8 | 1387169 | 0.97 | 1869 | 0.01 | 157066 | 0.32 | 1228234 | 2.50 |
| -8~-6 | 1454319 | 1.01 | 24085 | 0.05 | 216429 | 0.44 | 1213805 | 2.47 |
| -6~-4 | 4130606 | 2.88 | 256116 | 0.57 | 890150 | 1.81 | 2984340 | 6.07 |
| -4~-2 | 13657159 | 9.52 | 2128724 | 4.72 | 4939218 | 10.04 | 6589217 | 13.39 |
| -2-0 | 53574484 | 37.33 | 20495282 | 45.45 | 20455305 | 41.58 | 12623897 | 25.66 |
| 0~2 | 49899015 | 34.78 | 1907443 | 42.30 | 1728198 | 35.13 | 1354259 | 27.53 |





| | | | | | 8 | | 5 | | 2 | |
|---|---|---|---|---|---|---|---|---|---|---|
| 2~4 | 12292246 | 8.57 | 2665260 | 5.91 | 3926930 | 7.98 | 5700056 | 11.59 |
| 4~6 | 3834196 | 2.67 | 394775 | 0.87 | 905303 | 1.84 | 2534118 | 5.15 |
| 6~8 | 1543671 | 1.07 | 49438 | 0.11 | 242608 | 0.49 | 1251625 | 2.54 |
| >8 | 1715935 | 1.20 | 6493 | 0.01 | 181166 | 0.37 | 1528276 | 3.10 |
| 总计 | 143488800 | 100.00 | 45096480 | 100.00 | 49196160 | 100.00 | 49196160 | 100.00 |

**Table 2 Statistics of GIM grid TEC difference in 2021**

| Latitude range | Global | | High-latitude | | Mid-latitude | | Low-latitude | |
|---|---|---|---|---|---|---|---|---|
| TECu | Number | Percentage | Number | Percentage | TECu | Number | Percentage | Number |
| <-8 | 54383 | 0.02 | 0 | 0.000 | 1753 | 0.002 | 52630 | 0.06 |
| -8~-6 | 210410 | 0.08 | 4 | 0.000 | 12502 | 0.013 | 197904 | 0.21 |
| -6~-4 | 1019780 | 0.37 | 508 | 0.001 | 86405 | 0.092 | 932867 | 0.99 |
| -4~-2 | 7532261 | 2.74 | 134842 | 0.156 | 1243794 | 1.318 | 6153625 | 6.52 |
| -2~0 | 138278764 | 50.25 | 47008485 | 54.354 | 51650348 | 54.744 | 39619931 | 41.99 |
| 0~2 | 119542460 | 43.44 | 39143211 | 45.259 | 40082139 | 42.483 | 40317110 | 42.73 |
| 2~4 | 7035964 | 2.56 | 197936 | 0.228 | 1161618 | 1.231 | 5676410 | 6.02 |
| 4~6 | 1180953 | 0.43 | 1390 | 0.002 | 94836 | 0.101 | 1084727 | 1.15 |
| 6~8 | 254733 | 0.09 | 23 | 0.000 | 13363 | 0.014 | 241347 | 0.25 |
| >8 | 74292 | 0.03 | 1 | 0.000 | 2042 | 0.002 | 72249 | 0.08 |
| 总计 | 275184000 | 100.00 | 86486400 | 100.00 | 94348800 | 100.00 | 94348800 | 100.00 |

**Table 3 Statistics of GIM grid TEC difference of adjacent time in 2014**

| Latitude range | Global | | High-latitude | | Mid-latitude | | Low-latitude | |
|---|---|---|---|---|---|---|---|---|
| TECu | Number | Percentage | Number | Percentage | TECu | Number | Percentage | Number |
| <-8 | 10479090 | 11.87 | 893714 | 3.22 | 2548438 | 8.42 | 7036938 | 23.24 |
| -8~-6 | 4385966 | 4.97 | 886028 | 3.19 | 1546912 | 5.11 | 1953026 | 6.45 |
| -6~-4 | 6551684 | 7.42 | 1788660 | 6.45 | 2417978 | 7.99 | 2345046 | 7.75 |
| -4~-2 | 10320248 | 11.69 | 3705116 | 13.35 | 3886818 | 12.84 | 2728314 | 9.01 |
| -2~0 | 15925764 | 18.04 | 6864894 | 24.74 | 6113690 | 20.19 | 2947180 | 9.74 |
| 0~2 | 13388328 | 15.16 | 6527630 | 23.52 | 4811838 | 15.89 | 2048860 | 6.77 |
| 2~4 | 7645186 | 8.66 | 3620370 | 13.05 | 2634520 | 8.70 | 1390296 | 4.59 |
| 4~6 | 4803444 | 5.44 | 1741402 | 6.27 | 1824656 | 6.03 | 1237386 | 4.09 |
| 6~8 | 3425504 | 3.88 | 881570 | 3.18 | 1371966 | 4.53 | 1171968 | 3.87 |



| | Global | | High-latitude | | Mid-latitude | | Low-latitude | |
|---|---|---|---|---|---|---|---|---|
| >8 | 11375586 | 12.87 | 842296 | 3.03 | 3117744 | 10.30 | 7415546 | 24.49 |
| 总计 | 88300800 | 100.00 | 27751680 | 100.00 | 30274560 | 100.00 | 30274560 | 100.00 |

**Table 4 Statistics of GIM grid TEC difference of adjacent time in 2021**

| Latitude range | Global | | High-latitude | | Mid-latitude | | Low-latitude | |
|---|---|---|---|---|---|---|---|---|
| TECu | Number | Percentage | Number | Percentage | TECu | Number | Percentage | Number |
| <-8 | 541996 | 0.30 | 640 | 0.00 | 12256 | 0.02 | 529100 | 0.88 |
| -8~-6 | 1143808 | 0.65 | 2226 | 0.00 | 47338 | 0.08 | 1094244 | 1.81 |
| -6~-4 | 3337486 | 1.90 | 28196 | 0.05 | 291922 | 0.48 | 3017368 | 5.00 |
| -4~-2 | 11632594 | 6.61 | 806252 | 1.46 | 3157664 | 5.23 | 7668678 | 12.70 |
| -2~0 | 81152580 | 46.08 | 28959210 | 52.32 | 29582524 | 48.99 | 22610846 | 37.45 |
| 0~2 | 59890090 | 34.01 | 24852622 | 44.90 | 23518398 | 38.95 | 11519070 | 19.08 |
| 2~4 | 12211858 | 6.93 | 685080 | 1.24 | 3495616 | 5.79 | 8031162 | 13.30 |
| 4~6 | 4695480 | 2.67 | 15980 | 0.03 | 252478 | 0.42 | 4427022 | 7.33 |
| 6~8 | 1256092 | 0.70 | 1046 | 0.00 | 22238 | 0.04 | 1232808 | 2.04 |
| >8 | 255776 | 0.15 | 44 | 0.00 | 2798 | 0.00 | 252934 | 0.41 |
| 总计 | 176117760 | 100.00 | 55351296 | 100.00 | 60383232 | 100.00 | 60383232 | 100.00 |


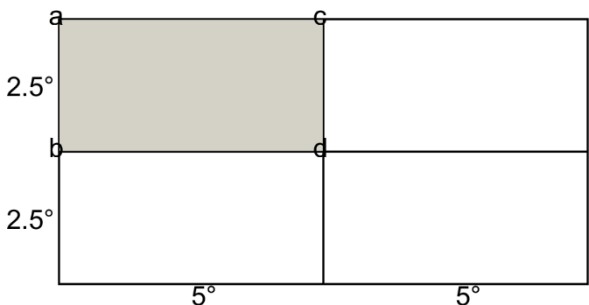

**Figure 1 The GIM grids diagram**

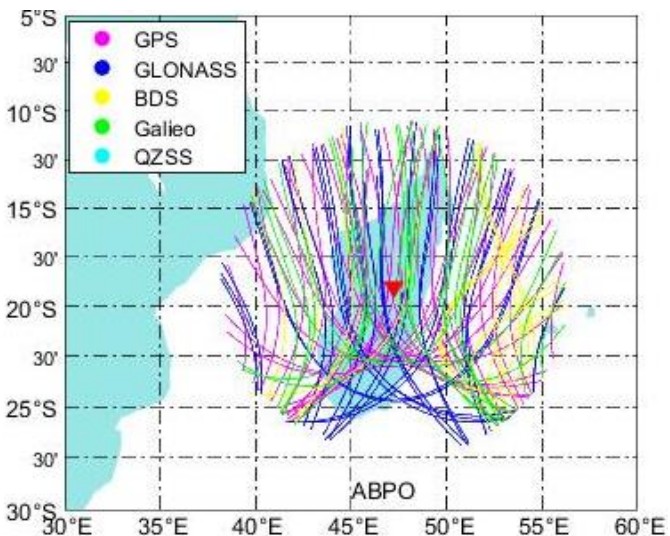

**Figure 2 Distribution of puncture points in ABPO (Different colors show the path of the puncture point formed by the observations of different satellites, and the red triangle indicates the location of the station)**

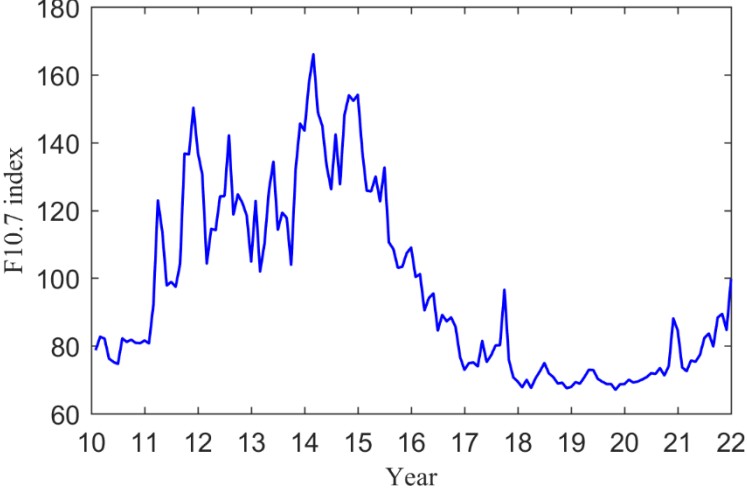

**Figure 3 Average monthly F10.7 index from 2010 to 2021**

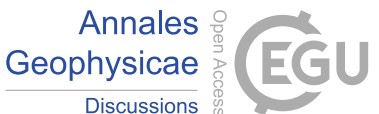

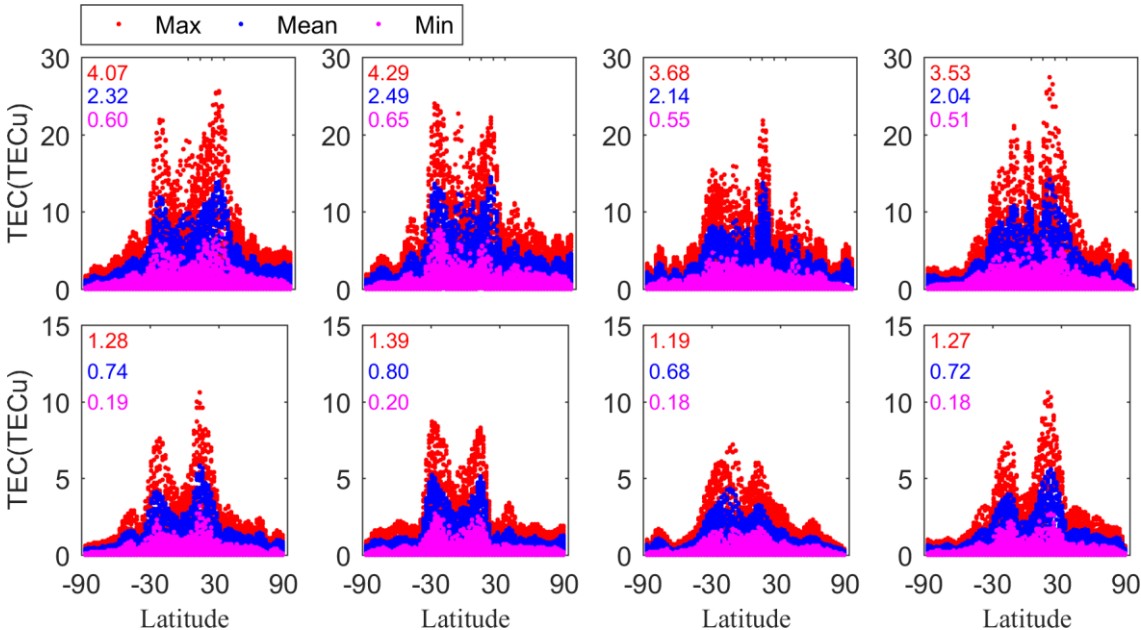

**Figure 4 Variation of GIM grid TEC difference in latitude (The first and second rows denote the results for 2014 and 2021, respectively, and the first to fourth columns represent the results for the four moments, respectively)**

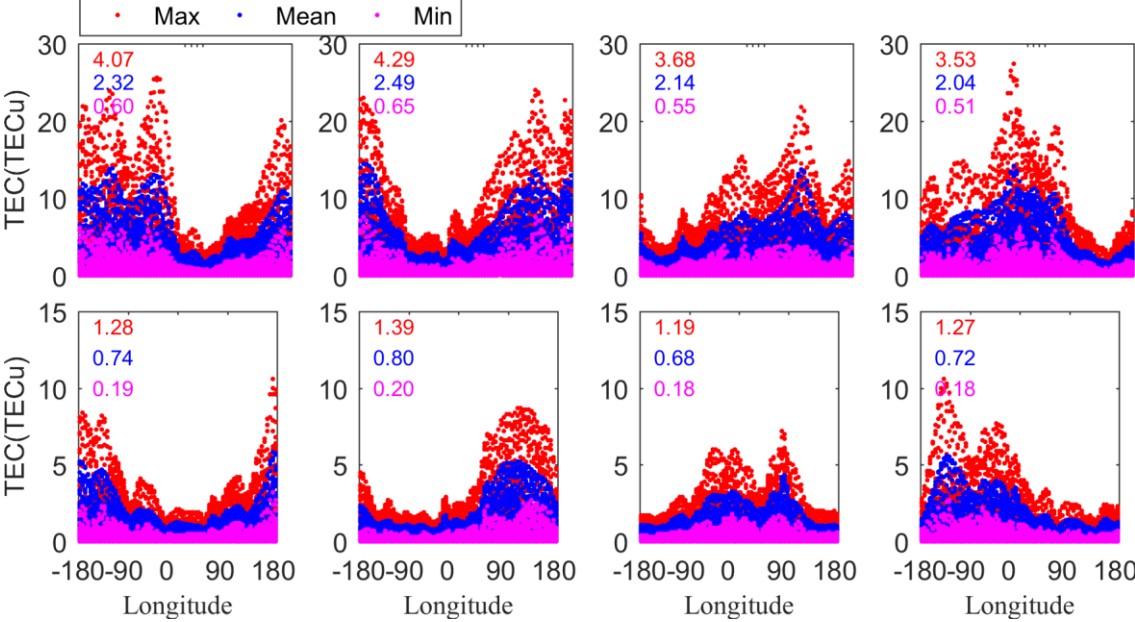

**Figure 5 Variation of GIM grid TEC difference in longitude (The first and second rows denote the results for 2014 and 2021, respectively, and the first to fourth columns represent the results for the four moments of the day, 02, 08, 14 and 20, respectively)**





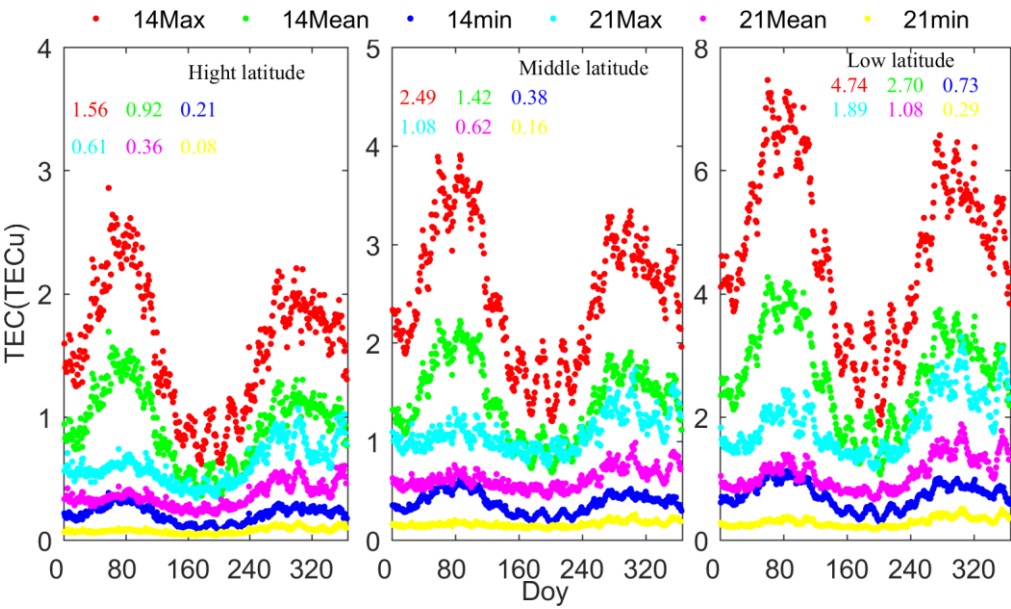

**Figure 6 Annual variation of grid TEC difference in GIMs**

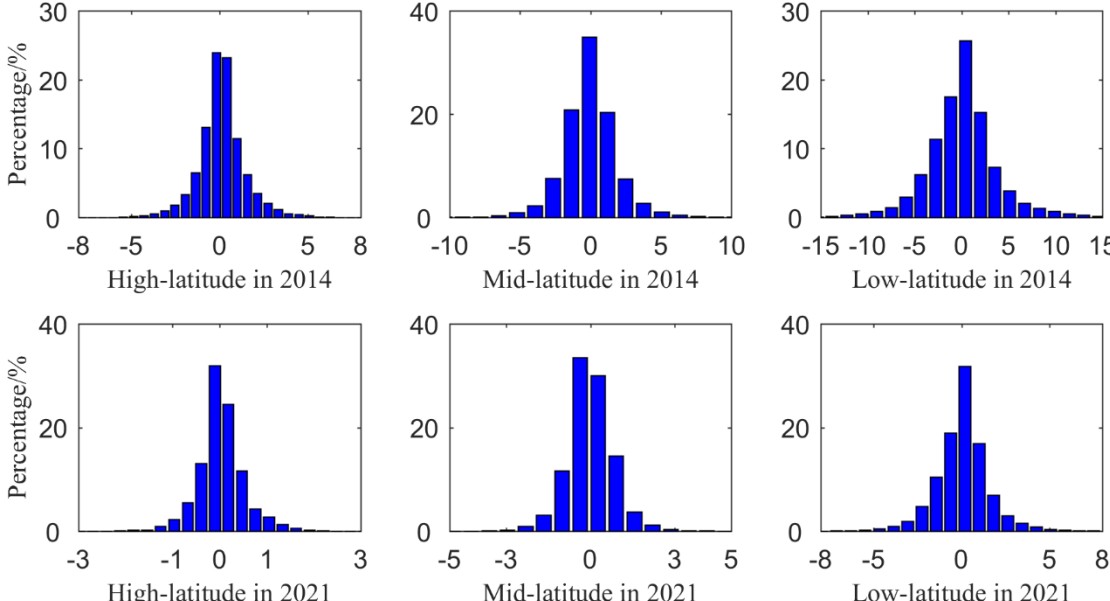

**Figure 7 Histogram of grid Tec difference**





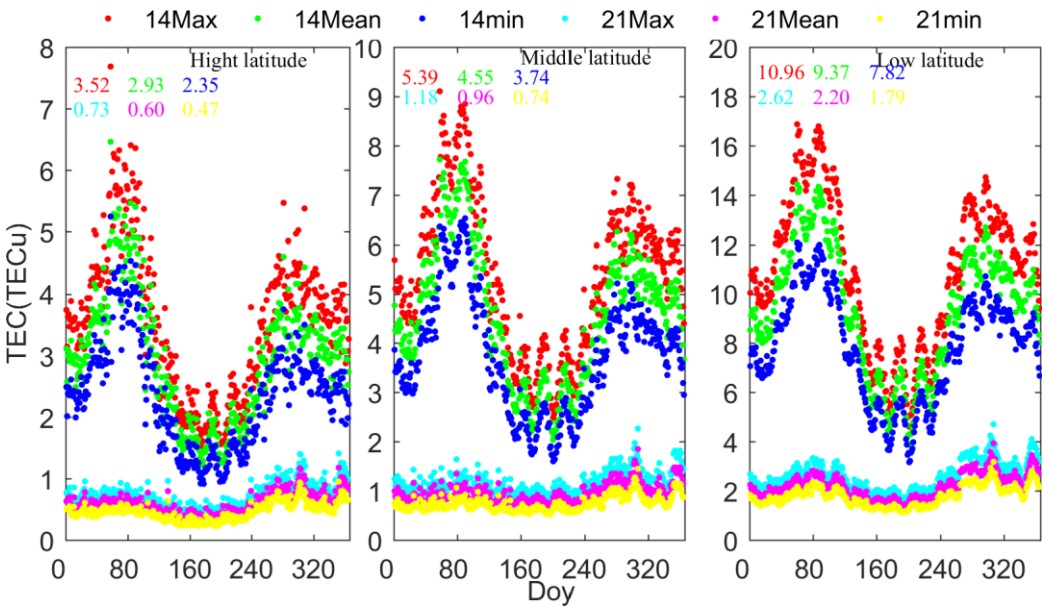

**Figure 8 Annual variation of grid TEC difference in GIMs**

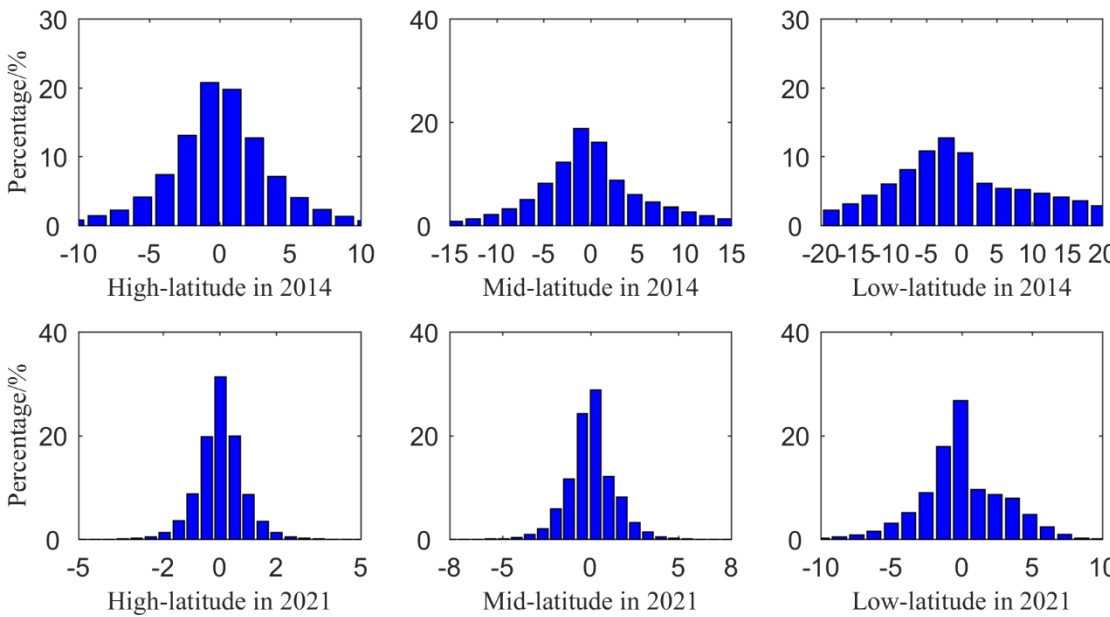


**Figure 9 Histogram of grid Tec difference**