# Peer review of "Characteristic Analysis of the Differences between TEC Values in GIM Grids"

_Annales Geophysicae, 2023_

## Author Comment (AC1)

Dear Reviewer:

Thanks very much for your comments. These comments were all valuable and very helpful for revising and improving our paper. In the revised manuscript, we have carefully revised it. The following is a point-to-point response to the comments

Thank you very much!

This work is very interesting and deals with something that many of us do not take into account when estimating variables which are obtained from an interpolation of nearby points. So, I consider it can be accepted for publication in this journal, but after additional analysis.
In my understanding, the authors aim to quantify the interpolation error when estimating the value of TEC at a point on the map based on the four nearest values from the GIM grid, and also when we analyze variation in time. The topic is interesting, but the conclusions they reach, which I think should be expected based on the well known spatial and time TEC variation, are not well interpreted. This could be done using the TEC data itself. I would expect, for example, a comparison of the variability of differences between the values of the grid points, with the actual TEC values themselves.

**Response:** **Thank you very much for your encouragement and comments. This paper is mainly based on the background, firstly, GIM is often used to obtain TEC values through grid interpolation, and secondly, there are few studies on TEC change characteristics within the GIM grid. Thus, we count the difference between the four-grid points TEC of each grid for the grid of GIM, and then analyze their change characteristics in space and time. The main finding of this analysis is that the grid TEC difference is small for most GIM grids, especially in the mid-high latitudes of low solar years. This means that the extraction methods and processes can be simplified when TEC within these GIM grids is needed.**

**We agree that a comparison of the variability of differences between the values of the grid points, with the actual TEC values themselves can be done. We will use real TEC data to analyze the error of interpolation points in the next step based on the analysis in this paper.**

In the case of the time variability analysis, in general, it can be expected that a parameter of a variable would be larger if the variable itself is larger. Therefore, given that TEC values are higher during high solar activity, I would logically expect that the temporal variation between consecutive times would also be greater, for example. When analyzing the latitude variation of the difference between grid points in a gird, the peaks in Figure 4 are similar to the peaks of the Equatorial Ionization Anomaly of the F2 region, which are also the region of greatest TEC spatial gradient. This could be checked by doing a plot of TEC in terms of latitude and longitude. That is a map. So, if this is the case this would be an explanation for greater difference in consecutive grid points.

**Response:** Yes, since the greater variation in ionospheric TEC values during high solar activity years, the difference in TEC values at grid points may also be greater. Yes, the peaks in Figure 4 are similar to the peaks of the Equatorial Ionization Anomaly of the F2 region. But it's not exactly the same. In Figure 4, the highest value is shown near 30° north-south latitude and decreases near the equator, with relatively small values appearing between them. Two peaks appear of figure 4 in near 30° north-south latitude.

The peaks shown in Figure 6 occur in equinox, which may be due to equinoctial maximum in TEC annual variation. This can be checked from the TEC values from where this figure was obtained also.

**Response:** Yes, it's like this.

The paragraph explaining Table 1 I consider it is obvious, and can be deduced from just seeing a TEC map in terms of longitude and latitude (a map, again). The high values are logical at equatorial and low latitudes, since in this region foF2 and TEC present the larger latitudinal gradients due to the Equatorial Ionization Anomaly. Also it is the region with most ionization disturbances. (I mention this same argument in a previous paragraph regarding Figure 4). May be I am wrong with this reasoning, but the authors can perfectly check this, and probably argue another reason for the behavior observed.

**Response:** Yes, some of the results in Table 1 can be inferred from the figure. However, the main purpose of Table 1 is to display the distribution frequency of the difference size between grids, specific difference values can be obtained.

High latitudes, close to the auroral zones, can also be regions with large ionization disturbances, but for a different reason. Here we have energetic particle fluxes that cause ionization perturbations.

**Response:** Yes, it's like this

The paragraph explaining Table 2 results (line 175-180), in my opinion is quite expected also. Low solar activity level implies low TEC value, and probably lower differences in absolute values (assuming the same pattern of variation along the solar cycle).

**Response:** The purpose of displaying Table 2 is also consistent with Table 1

In summary, I think that the behavior you observe in the variation of grid points (in space and time) are marked by the TEC variation in space and time. Greater TEC values implies greater time variation in grid points. And greater TEC values' spatial gradient (as it occurs in low and equatorial latitudes) implies greater differences between grid points in a given grid.
I suggest the authors to verify if this is the case.

**Response:** Yes, we agree with what you have mentioned above, the grid TEC difference is influenced by TEC variation. This paper mainly analyzes TEC variation characteristics within the grid. The main finding of this analysis is that the grid TEC differences are small for most GIM grids, especially in the mid-high latitudes of low solar years. This means that relevant extraction methods and processes can be simplified when TEC within these GIM grids is needed.

Minor comments.
Line 79: "It can be seen that It can be seen that the puncture points ..." delete one "It can be seen. That is should be "It can be seen that the puncture points ..."
Line 141: I think that "Like Figure 5, ..." should be "Like Figure 4, ..."
In line 142, you mention "... which is different from Figure 3." I think you wanted to say Figure 4. Since Figure 3 shows F10.7 variability.
In line 152: " This trend is the same as the trend of F10.7 in 2014 in Figure 1.". You probably mean " This trend is the same as the trend of F10.7 in 2014 in Figure 3." Please check.
Line 165: I think that "... depicted in Figure 5." should be "... depicted in Figure 7." Please check.

**Response:**
We have revised these in the revised manuscript.

We appreciate for your warm work earnestly, and hope that the correction will meet with approval. Once again, thank you very much for your comments and suggestions.

---

## Author Comment (AC2)

Dear Reviewer:

Thanks very much for your comments. These comments were all valuable and very helpful for revising and improving our paper. In the revised manuscript, we have carefully revised it. The following is a point-to-point response to the comments

Thank you very much!

This paper proposes the grid TEC difference to analyze TEC variation characteristics within the grid. This work is very interesting in my option. It can be accepted after appropriate revisions

Minor comments.

1. Some tables in the manuscript have obvious mistakes, such as Chinese words should be replaced with English words

**Response:**

We have made corrections in the revised manuscript.

2. There are errors in the reference of the figure in the analysis section., as like line 141-165. Please carefully check.

**Response:**

We have revised it in the revised manuscript.

3. The format of the table needs to be adjusted to better display relevant data.

**Response:**

We have made corrections in the revised manuscript.

4. Some Figures is not clear, it is difficult to distinguish what each picture represents. Please check the manuscript carefully.

**Response:**

We have made corrections in the revised manuscript.

5. In line 152, you mention" ······F10.7 in 2014 in Figure 1.". You probably mean " This trend is the same as the trend of F10.7 in 2014 in Figure 3." Please check.

**Response:**

We have revised it in the revised manuscript.

We appreciate for your warm work earnestly, and hope that the correction will meet with approval. Once again, thank you very much for your comments and suggestions.

---

## Author Response (AR2)

Dear Reviewer:

Thanks very much for your comments. These comments were all valuable and very helpful for revising and improving our paper. In the revised manuscript, we have carefully revised it. The following is a point-to-point response to the comments

Thank you very much!

* Leave a space after a semicolon (check list of references between brackets). Check all the paper please.

**Response:**

We have revised it in the revised manuscript.

*Line 42: "Since the ionosphere is influenced by solar activity, its system state and variation are complicated, but it is generally believed that the active level of the ionosphere is related to solar activity". I would delete the last part of this sentence, because it is not just a matter of "belief". The ionosphere is affected by solar activity. So it should be "Since the ionosphere is influenced by solar activity, its system state and variation are complicated."

**Response:**

We have revised it in the revised manuscript.

* Line 45: " time(Tariq" there should be a space after "time". That is, it should be " time (Tariq"

Check please in other parts. There should be a space between a word and a bracket when you are opening it, that is when you use "(".

**Response:**

We have revised it in the revised manuscript.

* Line 79: " (GNSS tracking station)," should end in a point. That is " (GNSS tracking station)."

**Response:**

We have revised it in the revised manuscript.

* Line 87: "In this paper, the grid TEC difference is proposed aims to analyze the TEC variation characteristics within the grid." Something is wrong in this sentence.

Maybe it should be "In this paper, the grid TEC difference is proposed to analyze the TEC variation characteristics within the grid." Please, check.

**Response:**

We have revised it in the revised manuscript.

We appreciate for your warm work earnestly, and hope that the correction will meet with approval. Once again, thank you very much for your comments and suggestions.